# Universal superconducting precursor in three classes of unconventional superconductors

D. Pelc [1], Z. Anderson[1], B. Yu[1], C. Leighton [2] & M. Greven[1]

A pivotal challenge posed by unconventional superconductors is to unravel how superconductivity emerges upon cooling from the generally complex normal state. Here, we use nonlinear magnetic response, a probe that is uniquely sensitive to the superconducting precursor, to uncover remarkable universal behaviour in three distinct classes of oxide superconductors: strontium titanate, strontium ruthenate, and the cuprate high-$T_c$ materials. We find unusual exponential temperature dependence of the diamagnetic response above the transition temperature $T_c$, with a characteristic temperature scale that strongly varies with $T_c$. We correlate this scale with the sensitivity of $T_c$ to local stress and show that it is influenced by intentionally-induced structural disorder. The universal behaviour is therefore caused by intrinsic, self-organized structural inhomogeneity, inherent to the oxides' perovskite-based structure. The prevalence of such inhomogeneity has far-reaching implications for the interpretation of electronic properties of perovskite-related oxides in general.

[1] School of Physics and Astronomy, University of Minnesota, Minneapolis MN-55455, USA. [2] Department of Chemical Engineering and Materials Science, University of Minnesota, Minneapolis MN-55455, USA. Correspondence and requests for materials should be addressed to D.P. (email: dpelc@umn.edu) or to M.G. (email: greven@umn.edu)

 

Some of the most prominent unconventional superconductors are based on the perovskite crystal structure. This includes the materials we study here: $SrTiO_3$ (STO), $Sr_2RuO_4$ (SRO), and the cuprates $La_{2-x}Sr_xCuO_4$ (LSCO) and $HgBa_2CuO_{4+\delta}$ (Hg1201). These oxides are of tremendous scientific interest: superconductivity in doped STO occurs at some of the lowest known charge carrier densities and, more than five decades after its discovery[1], there has been a recent upsurge of research activity[2–4]; SRO is a candidate for unconventional spin-triplet superconductivity[5]; LSCO and Hg1201 are archetypal cuprate high-$T_c$ superconductors, arguably the most widely investigated class of quantum materials[6]. These materials have widely different values of $T_c$, and, although their respective superconducting pairing mechanisms remain unknown, they are thought to differ as well. They do, however, share a common structural motif—the perovskite metal-oxygen octahedron—which is known to be susceptible to various deformations[7,8]. One of the salient open questions in the physics of unconventional superconductors is the nature of the superconducting precursor above $T_c$, where macroscopic superconductivity is absent, but traces of pairing remain observable. Experimental studies of the precursor can, in principle, provide pivotal information on the superconductivity, yet are hampered by the difficulty of distinguishing tiny superconducting signatures from normal state properties.

We probe the superconducting precursor regime above $T_c$ by measuring nonlinear magnetic response. The magnetic response of a material to an external magnetic field, beyond the linear approximation, may be written as

$$M = \chi_1 H + \chi_3 H^3 + \dots \qquad (1)$$

where the $\chi_i$ are the susceptibility tensors, $M$ is the magnetization, and $H$ is the applied magnetic field. We assume that time-reversal symmetry is obeyed, and hence do not consider even-power terms. Superconducting transitions are clearly seen in linear response because of the diamagnetism associated with the superconducting state, but above $T_c$ the properties of the non-superconducting normal state also contribute to $\chi_1$. However, the third-order diamagnetic response is virtually zero in the absence of superconducting correlations, and is thus uniquely sensitive to the superconducting precursor. We show here that, surprisingly, the three different classes of oxide unconventional superconductors share a common precursor regime, with a likely origin in inherent inhomogeneity related to the common perovskite-based structure.

## Results

**Nonlinear response measurements**. Our experiments are performed with an alternating field, $H = H_0 \cos(\omega t)$, applied via an excitation coil, and the response is detected with a pair of pickup coils at frequencies $\omega$ (linear response) and $3\omega$ (nonlinear response). A typical excitation amplitude is $H_0 = 1\,Oe$. Phase-sensitive detection enables frequency selectivity, with extremely high sensitivity in the kHz range (see Methods). The present experiment should be contrasted with previous radio-frequency measurements of cuprate nonlinear response, which were analysed in terms of conductivity owing to the much larger induced electric fields[9]. We discuss all results here in terms of the raw third-harmonic signal, $M_3$.

Remarkably, we find the same behaviour of the superconducting precursor in all four studied oxides (Fig. 1). The data clearly show exponential temperature dependence that spans at least two orders of magnitude in $M_3$, with the temperature range limited only by detection noise. Furthermore, there is no appreciable change with excitation frequency. The characteristic scale $\Xi$ of the exponential decrease is obtained from fits to $M_3 \sim e^{-T/\Xi}$. To highlight the universality and robustness of the precursor behaviour, the data for different compounds are shown to collapse on a master curve upon scaling the temperature with $\Xi$ (Fig. 2a), which is consistent with simple exponential decay in a wide range of $M_3$. As shown in Fig. 2b, $\Xi$ clearly increases with $T_c$, but at a significantly sublinear rate. The values should be viewed with some caution, however, as for STO, LSCO, and Hg1201, $T_c$ depends on the carrier-doping level. We work here with nearly

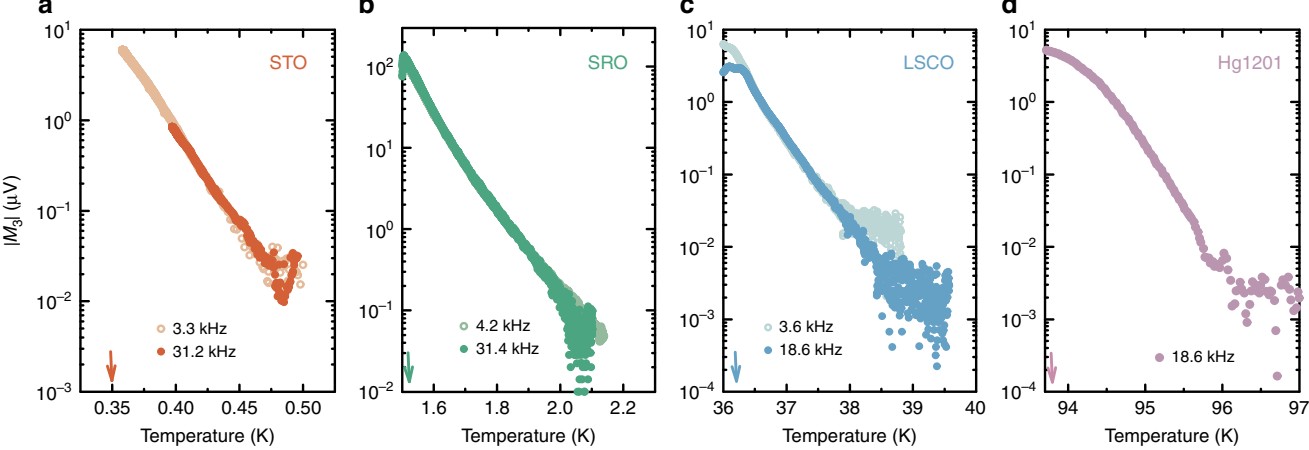

**Fig. 1** Nonlinear response of four perovskite-based superconductors above $T_c$. **a** Strontium titanate $SrTiO_3$ (STO) doped with 1 at.% of Nb, with a $T_c$ of 350 mK. **b** Strontium ruthenate $Sr_2RuO_4$ (SRO), with a $T_c$ of 1.51 K. **c** Slightly underdoped lanthanum strontium cuprate $La_{2-x}Sr_xCuO_4$ (LSCO), with a $T_c$ of ~ 36 K (Sr doping level of 14%). **d** Slightly underdoped mercury barium cuprate $HgBa_2CuO_{4+\delta}$ (Hg1201), with a $T_c$ of ~ 94 K. Measurements on four other Hg1201 samples are presented in Supplementary Note 3 and Supplementary Fig. 4, with essentially the same results. Arrows indicate the respective $T_c$ values. For the three lamellar compounds **b–d**, the magnetic field is perpendicular to the transition-metal-oxygen planes, whereas for STO it is along one of the high-symmetry cubic directions. The four oxides show an exponential tail that stretches over at least two orders of magnitude, and with a slope that is approximately independent of excitation frequency. The low-frequency results are raw data, whereas the high-frequency results were multiplied by a constant to coincide with the low-frequency measurements. The constants depend on the setup and sample penetration depth, and are 0.7, 5, and 0.2 for STO, SRO, and LSCO, respectively (Hg1201 is measured at only one frequency)

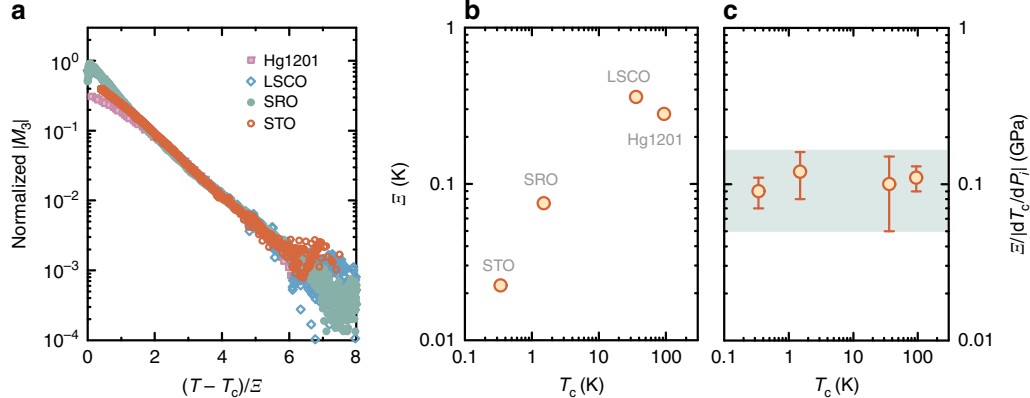

**Fig. 2** Characteristic pre-pairing scale. **a** Scaling plot of the raw data from Fig. 1, demonstrating the universality of the precursor response. The temperature axis is scaled by the slopes of the exponential dependences, $\Xi$, and the curves are multiplied by constant factors to obtain collapse on the vertical axis. High-frequency data from Fig. 1 are used, except for STO where we combined both data sets. **b** The slopes $\Xi$ plotted against the $T_c$ values of the studied compounds. Errors are obtained from exponential fits and are smaller than the symbol size. The exponential fits show high fidelity, as seen in the low values of the reduced sum of residuals (Table 1). **c** $\Xi$ normalized to the derivative of $T_c$ with regard to uniaxial stress (Table 1). For STO, the derivatives $dT_c/P_a$, $dT_c/P_b$, and $dT_c/P_c$ with respect to the three principal crystallographic directions are similar owing to the nearly cubic structure. For SRO, LSCO, and Hg1201, we take values of $dT_c/dP_i$ with the pressure applied within the $RuO_2$ and $CuO_2$ planes. For SRO, the dependence of $T_c$ on $P_a$ is nonlinear, and we take the average slope between $T_c = 1.5$ K and $T_c = 2$ K. The nonlinearity may be the cause of the slight curvature seen in Fig. 1b. The turquoise band indicates the estimated uncertainty range, and the error bars are obtained directly from the uncertainties of $dT_c/dP_i$ (see Table 1).

---

**Table 1 Characteristic quantities for the oxide superconductors**

| Compound | $\Xi$ (mK) (fit range) | Reduced $\chi^2$ | $|dT_c / dP_i|$ (K/GPa) | Coherence length (nm) |
|---|---|---|---|---|
| STO | 22.4 ± 0.1 (0.36 K - 0.47 K) | $2.5 \cdot 10^{-3}$ | 0.25 ± 0.05 (ref. [24]) | ~30 (ref. [31]) |
| SRO | 75.0 ± 0.6 (1.55 K–2 K) | $6.5 \cdot 10^{-3}$ | 0.6 ± 0.2 (ref. [25]) | ~75 (ref. [5]) |
| LSCO | 358 ± 1 (36.5 K–38.5 K) | $4.2 \cdot 10^{-3}$ | 3.7 ± 1.3 (ref. [26]) | ~2 (ref. [32]) |
| Hg1201 | 272 ± 3 (94.5 K–96.5 K) | $7.8 \cdot 10^{-3}$ | 2.3 ± 0.3 (ref. [26]) | ~2 (ref. [32]) |

The sums of the squared residuals normalized to the number of experimental points (reduced $\chi^2$) is given for linear fits to the scaled log $M_3$ curves (Fig. 2a) so as to be directly comparable. For SRO, LSCO, and Hg1201, the average in-plane values of the derivatives are used; the large error margin for LSCO reflects the fact that the two principal in-plane values differ by a factor of 2. For SRO, the dependence of $T_c$ on $P_a$ is strongly nonlinear, and we take the average slope between $T_c = 1.5$ K and $T_c = 2$ K by simply fitting a linear function to data for two samples in ref. [25]

---

optimally-doped crystals (with the highest $T_c$) as representative examples.

Regardless of interpretation, the simple universal behaviour is remarkable. First, the extended exponential tail is distinctly different from what is expected and observed in conventional superconductors. We measure $M_3(T)$ for various conventional superconductors, and indeed find that the nonlinear response decays quickly below the noise level and exhibits power-law behaviour consistent with Ginzburg-Landau theory and its extensions[10,11] (see Supplementary Note 1 and Supplementary Fig. 1). Second, the oxides feature varied transition-metal ions, a wide range of optimal $T_c$ values, and pairing mechanisms that are thought to be quite different. STO superconductivity may be phonon[2]- or plasmon[3]-mediated, or related to ferroelectric criticality[4], SRO is a possible triplet superconductor with a potentially important role played by ferromagnetic fluctuations[5], and LSCO and Hg1201 are members of the cuprate high-$T_c$ family known to exhibit unusual pseudogap phenomena[6]. Moreover, the crystal (and electronic) structures differ as well. STO is a cubic perovskite that transforms to tetragonally distorted on cooling, whereas SRO, LSCO, and Hg1201 belong to the perovskite-related, layered Ruddlesden–Popper series. However, all compounds feature the common perovskite transition-metal-oxygen octahedron and the associated general tendency toward structural deformation[7,8]. Both STO and LSCO exhibit a well-defined symmetry-lowering structural transition on cooling[12,13], Hg1201 is nominally tetragonal, but with a sizable distribution of local crystal fields[14], whereas SRO is close to an octahedral rotation instability[15].

**The role of structural inhomogeneity**. The observed exponential behaviour is highly unusual, and similar features are scarce in condensed matter physics. Exponential tails can appear in an ordered system with a coupling of the order parameter to local inhomogeneity; rare regions with exceptionally high local ordering temperatures then cause the tails[16]. A similar effect occurs in the dynamics of electrons in a random lattice potential, giving rise to exponential tails in the density of states of disordered semiconductors[17,18]. The same physical arguments can be adapted to superconductors, if the inhomogeneity occurs at length scales comparable to the superconducting coherence length[19]. In the cuprates, the exponential temperature dependence of the precursor superconductivity was recently characterized using several complementary techniques — linear[20] and nonlinear[9] conductivity, as well as torque magnetization[21]— and it was found that a simple model based on spatial $T_c$ inhomogeneity agrees well with the data[9,20]. By extension, this indicates that similar distributions universally exist in perovskite-based superconductors. Furthermore, as the only certain common feature shared by the distinct superconductors studied here is structure, the underlying origin of the electronic inhomogeneity must be related to structural inhomogeneity. All studied compounds are oxides, and oxygen non-stoichiometry might play a role. However, we note that the inhomogeneity is at most indirectly related to

doping-related point defects, such as oxygen vacancies. Namely, the different oxides have widely varying levels of defect concentrations: LSCO is substitutionally doped, and the Sr substituents cause strong local field variations that depend on the doping level; Hg1201 is doped with oxygen interstitials relatively far from the conducting $CuO_2$ planes, so their effects are less pronounced; the dopant concentration in STO is quite small compared with cuprates; and SRO is nominally stoichiometric, with minimal point disorder (although ruthenium inclusions can influence the results; see Supplementary Note 2 and Supplementary Fig. 3). This suggests that the inhomogeneity is likely caused by intrinsic local stress accommodation that leads to atomic displacements, with the possibility that oxygen defects correlate with lattice deformations. In metals and alloys, structural transitions that are displacive (i.e., non-diffusive) and symmetry-lowering are known as martensitic transitions[23]; they are accompanied by local deformations across multiple length scales, which can exist even if there is no macroscopic transition[8]. Analogous deformations in the perovskite-based oxides could be the underlying cause of the extended precursor regime that we observe.

We test this hypothesis in two ways. First, we emphasize the following simple observation. In all studied oxides, $T_c$ quite strongly depends on lattice deformation. A straightforward means to quantify this is the derivative of $T_c$ with respect to uniaxial stress, $dT_c/dP_i$, where $P_i$ is the pressure applied in a given crystallographic direction $i$. If lattice deformation at the nanoscale (i.e., at a scale comparable to the superconducting coherence length) is the cause of inhomogeneity that gives rise to exponential behaviour, we would still expect $dT_c/dP_i$ to be a reasonable measure of the coupling between local $T_c$ and local (nanoscale) lattice strain. Therefore, the scale $\Xi$ should be related to $dT_c/dP_i$. Indeed, as shown in Fig. 2b, upon normalizing $\Xi$ by $dT_c/dP_i$ for the four different oxides[24–26] we obtain a nearly universal value. $dT_c/dP_i$ is negative for STO, but as we only use the derivative as a measure of coupling to lattice deformation, its absolute value is relevant for the determination of the $T_c$

inhomogeneity scale. We thus demonstrate that

$$\Xi \approx A \left| \frac{dT_c}{dP_i} \right|, \qquad (2)$$

where the constant $A \sim 100\,MPa$ plays the role of a characteristic stress scale, and may be interpreted as a measure of the intrinsic local stress distribution width. As the elastic moduli of the oxides[22,27] are on the order of 100 GPa, the associated lattice deformations amount to 0.1% of the lattice parameter, or $\sim 0.1$ pm. This estimate may be crude, but it serves to show that the relevant deformations at the atomic level are, on average, very small. Yet, they may be within reach of high-precision neutron scattering[28] and electron microscopy[29] techniques. Notably, the simple relation (2) is borne out in previous studies of the cuprates. It is known from thermodynamic studies[26] that $dT_c/dP_i$ is nearly universal in the cuprates, and a characteristic precursor scale similar to $\Xi$ was also found to be universal[9,20,21], in accordance with (2). Importantly, the oxides studied here have very different superconducting coherence lengths of $\sim 75$ nm (SRO)[5], $\sim 30$ nm (STO)[31] and 1–3 nm (cuprates)[32]. As $A$ appears to be universal, the relevant local strain should have a scale-free structure to cause the same effect on $T_c$.

A more direct test of the structural origin of the exponential tails is to study the influence of intentionally introduced structural inhomogeneity. An LSCO single crystal was thus subjected at room temperature to in-plane uniaxial pressure beyond the elastic regime (Fig. 3a, b), reaching stresses comparable to $A$. A relatively low yield stress is observed, followed by plastic deformation and hysteresis; such features are not uncommon in compressively deformed single crystal oxide ceramics[27], but to our knowledge have not been explored in the cuprates. The deformation must induce structural inhomogeneity (defects) to accommodate plasticity, beyond what is already present in the undeformed sample. We then compare nonlinear magnetization measurements on the same sample before and after applying 50 MPa pressure (Fig. 3c). The exponential slope changes dramatically, which demonstrates a

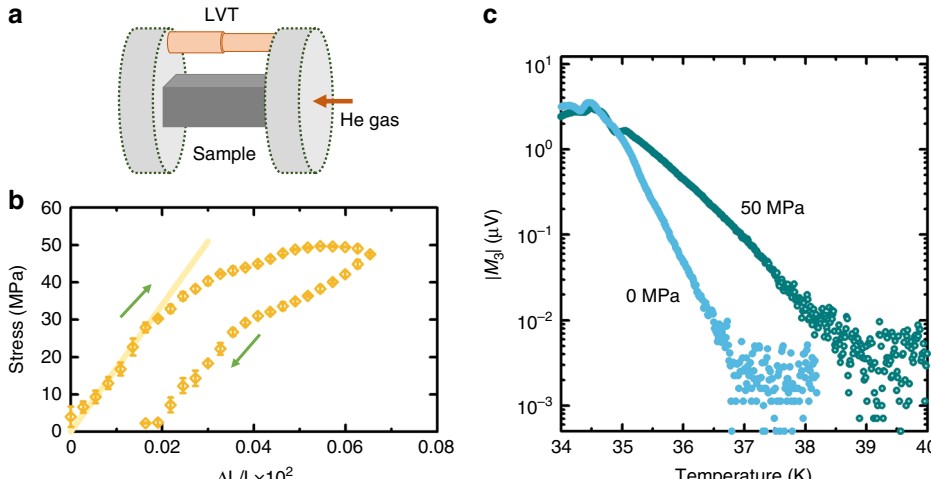

**Fig. 3** Influence of stress-induced inhomogeneity. **a** Schematic representation of the uniaxial pressure experiment, where a controlled force is applied to the sample with a helium gas piston (right disk), whereas the other side is static (left disk). The sample deformation is measured independently using a linear variable transformer (LVT). **b** Ambient temperature stress-strain diagram for a LSCO sample with Sr doping level slightly below 14% (from a different growth than the sample in Fig. 2). The deviation from linear behaviour and the hysteresis clearly show that the plastic regime is reached, thus inducing structural inhomogeneity (defects) to accommodate plasticity. Multiple pressure readings were acquired at each deformation, and error bars are one standard deviation from the mean. The line corresponds to a Young modulus of 170 GPa, in good agreement with the value obtained from ultrasonic measurements of the elastic tensor components, 181 GPa (ref. [22]). **c** Nonlinear magnetization measurements before (full circles) and after (empty circles) the application of stress at room temperature. The stress-induced inhomogeneity has a strong influence on the slope of the exponential temperature dependence, and thus establishes its structural origin. Note that the slope of the unstressed sample is the same as in Fig. 1, which demonstrates a high level of repeatability between different growths

clear relation to structural inhomogeneity. The ratio of the slopes in Fig. 3c is 1.7; if $\Xi$ is simply proportional to the local stress distribution width, this change is roughly what could be expected from the naive addition of 50 MPa of induced stress to 100 MPa of pre-existing internal stress. Remarkably, the applied stress creates regions with locally increased $T_c$ compared with the unstressed sample.

## Discussion

Our results have at least four far-reaching implications. First, we have uncovered an unanticipated common feature among perovskite-based superconductors, present regardless of the significant differences in their electronic properties. Second, the strong link between the characteristic temperature scale $\Xi$ and the coupling of $T_c$ to lattice strain suggests a common structural origin, rooted in the propensity for local deformation inherent to perovskites. Such effects have already been explored for some systems, especially the colossal magnetoresistive manganites[30], but should be relevant for perovskite-based materials in general, be they bulk superconductors, ferromagnets, ferroelectrics, or films/heterostructures[33]. This insight is already beginning to be applied to the analysis of experiments on cuprates[9,20,21,34,35]. In Supplementary Fig. 2, we show for SRO that the exponential temperature dependence possibly is also present in a rather different observable: the electronic specific heat. Third, we demonstrate that it is viable to manipulate the inhomogeneity using mechanical stress, creating regions of locally increased $T_c$. Such experiments open up a new avenue for the study of unconventional superconductors; they should be invaluable in elucidating the nature of the intrinsic inhomogeneity with structural probes, and pave the way toward stress-engineering of superconducting properties. Finally, the observation of a universal intrinsic stress $A$ suggests that the underlying local deformations are similar in bulk perovskite-related materials, at least at the scale of the superconducting coherence length. Since for the oxides considered here these lengths differ by more than an order of magnitude, it is plausible that the deformations are self-organized and scale-free to some extent. This is consistent with the suggestion that the interstitial oxygen atoms in the cuprate $La_2CuO_{4+\delta}$ form scale-invariant structures[36]. Our work indicates that such phenomena are a general property of perovskite-based materials, and it will be interesting to understand if they are prevalent in other unconventional superconductors.

## Methods

**Samples**. The bulk STO sample is a single crystal cut from a commercially available wafer doped with 1 at.% of Nb, with the structure, part-per-million-level purity[37], and normal state characterized in previous work[38]. The bulk SRO crystals, grown by the traveling-solvent floating-zone technique and characterized by X-ray diffraction, show a high degree of crystalline perfection. The high value of $T_c = 1.51$ K indicates that the crystal in Fig. 1 is in the clean limit, i.e., that the level of point disorder is minimal[5]. The LSCO samples are grown by the traveling-solvent floating-zone method and characterized with X-ray diffraction. The sample for which the data in Fig. 1 were obtained is also characterized with energy-dispersive X-ray spectroscopy, and the doping level $x = 0.14$ is found to be homogeneous across the sample. All LSCO samples are annealed post-growth in air at 800 °C for 100 h. The Hg1201 samples are grown using a previously described encapsulation method[39] and annealed in flowing oxygen gas for 1 month. The Pb and In samples are commercial 99.9% purity polycrystalline samples cut into rectangular bars, whereas the Nb sample is a commercially available single crystal. The $Sn_{97}Ag_3$ sample is a standard soldering alloy.

**Nonlinear magnetization**. In order to measure the third-order nonlinear susceptibility, two different experimental setups are used. For measurements below 2 K, we employ a $^3$He-recirculating refrigerator, whereas for measurements at higher temperatures we constructed a dedicated probe used within a Quantum Design, Inc., magnetic property measurement system (MPMS). The $^3$He setup uses a set of one excitation and two detection coils, with the detection coils connected in such a way as to cancel the excitation field. The excitation current is supplied by a Stanford Research DS360 ultra-low distortion generator, and for detection we use an EG&G 5302 lock-in amplifier with transformer input (below 10 kHz) and with direct pre-amp input (above 10 kHz). The additional distortion from the transformer and pre-amp is negligible in the frequency range of interest. The sample and coils are shielded by a double Pb can superconducting shield, and we estimate the residual field in the sample space to be on the order of 0.01 Oe. The MPMS probe only contains the two matched detection coils, whereas for excitation we use the built-in coil for AC susceptometry. The MPMS is otherwise only used for temperature control. Both excitation and detection in the MPMS setup are provided by a Signal Recovery 7265 lock-in amplifier; we do not use a transformer. The sample is shielded by an external mu-metal shield, with the superconducting magnet of the MPMS quenched (reset) at low field before the experiments. The residual field in this configuration is estimated to be no larger than 0.1 Oe. The excitation fields in all experiments are on the order of 1 Oe (without corrections for demagnetization factors).

**Uniaxial stress**. Uniaxial pressure is applied to the sample at 300 K with a cell that uses helium gas pressure to create a force on a piston which, in turn, compresses the sample. The design enables independent control and measurement of applied pressure and sample strain. Tungsten carbide composite blocks are used to transfer the force to the sample, and the sample deformation is measured with a linear variable transformer that consists of two concentric coils wound on ceramic formers attached to the piston and stationary end, respectively. The sample used in these experiments has a cross-section of 0.64 mm² and is 1.26 mm long; the cell force-He pressure ratio is 61 N/bar. Therefore, ~ 0.5 bar is needed to obtain the 50 MPa stress in the sample studied here.

## Data availability

The data sets generated during the current study are available from the corresponding authors on reasonable request.

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

## Acknowledgements

We thank W. Zimmermann for the donation of the ³He fridge used (and for invaluable advice on its operation), A.S. Gibbs, C.W. Hicks and A.P. Mackenzie for providing the $Sr_2RuO_4$ samples, N. Bielinski for help in preparing the LSCO samples, S. Griffitt for assistance in constructing the susceptometer probes, A. Najev and M. Lukas for assistance with the uniaxial pressure measurements, and N. Barišić and B. Shklovskii for discussions. The work at the University of Minnesota was funded by the Department of Energy through the University of Minnesota Center for Quantum Materials under DE-SC-0016371.

## Author contributions

D.P. and M.G. initiated the work. D.P. designed the probes and performed the nonlinear response measurements, with assistance from Z.A. and B.Y. Z.A. and B.Y. grew and characterized the Hg1201 and LSCO samples, respectively. C.L. provided and characterized the STO sample. D.P. carried out the data analysis. D.P. and M.G. wrote the manuscript, with input from all authors.

## Additional information

**Competing interests:** The authors declare no competing interests.

