## [Peer Review File · Nature Communications]

Reviewers' comments:

Reviewer #1 (Remarks to the Author):

The manuscript by D. Pelc et al. reports an universal diamagnetic signature observed above the superconducting transition temperature (T_c) of various transition-metal oxide superconductors, namely SrTiO₃ (STO), Sr₂RuO₄ (SRO), La_{1.86}Sr_{0.14}CuO₄ (LSCO) and HgBa₂CuO₄ (Hg1201). Measurements of the third-harmonic magnetic response (M_3) to an applied field (H) in the kHz regime, made on both as-grown and uniaxially-strained single crystals, showed a predominantly exponential dependence of M_3 on temperature (T), in contrast to a power-law behavior that is expected from Ginzburg-Landau theory for conventional superconductors.

This work has both technical novelty and crucial significance for the study of superconducting fluctuations above the bulk T_c in oxide superconductors, where the superconducting energy gap is known to vary locally due to both chemical and structural inhomogeneities, the latter being intrinsic to oxide perovskites.

First, the present experiment is technically distinct from prior measurement of nonlinear electromagnetic response made in terms of conductivity at radio frequencies (Ref. 8). To my best knowledge, the simultaneous measurement of M_3 and sample strain reported here are the first of its kind. Importantly, by also performing control experiments on the conventional superconductors V, Pb, SnAg and Nb, which expectedly show power-law behavior, the technique used and thus the unconventional behavior observed in STO, SRO, LSCO and Hg1201 are properly validated.

Second, the fact that the magnitude of M_3 vs. T varies with uniaxial strain, while the exponential behavior remains unchanged, crucially connects this unconventional behavior to the presence of structural inhomogeneities. The appearance of a common ratio, of the exponential temperature scale to $dT_c/d\epsilon$ (from Eqn. 2), is strong evidence for a universal pre-pairing scale (in Fig. 2b) rooted in local defects. These observations also suggest that strain engineering can be effectively used to manipulate local inhomogeneities as a tuning parameter of the superconducting fluctuations in these oxides.

In my opinion, the technical and scientific merits of this work amply justify its publication in Nature Communications. The manuscript is well written, the data clearly presented and the key results are of both general interest and timely importance. Listed below are some questions of curiosity that the authors may consider addressing, along with some constructive suggestions for further improving the manuscript:

1) Since the exponential behavior of M_3 vs. T is attributable to spatial inhomogeneity of the superconducting energy gap (Δ_0), it may be elucidating to plot the M_3 data vs. a "reduced energy" scale rather than a "reduced temperature" scale. Such a plot could perhaps allow all three panels of Fig.1 to collapse into one master plot, thus further strengthening the universality. In this regard, it is worth noting that the $2\Delta_0/k_B T_c$ ratio for cuprate superconductors tends to deviate from mean-field value, as T_c is increased (J. Wei et al., Phys. Rev. B 57, 3650 (1998)). It is also of interest to note the difference between "pairing" and "condensation" energies, with the latter reflecting T_c , in relation to the pseudogap phenomenon (G. Deutscher, Nature 397, 410 (1999)).

2) The strain hysteresis shown in Fig. 3 suggests that M_3 vs. T could also be hysteretic, as one may expect from irreversible inhomogeneities associated with the plastic deformation. It may be worthwhile to elaborate on this point in the manuscript. In particular, would the occurrence of the latter hysteresis provide further evidence for strain-induced inhomogeneities?

3) In Fig. 1a, it would be clarifying to indicate the orientations of the crystal and of the magnetic field. For the latter, it would also help to qualify whether and how demagnetization factors, due to sample shape and field orientation, may affect the M3 vs. T data.

4) Finally, the title of the manuscript could be made a little less abstract, particularly for the general reader who may not be familiar with the phenomenological complexities of precursor superconductivity in transition-metal perovskites.

Reviewer #2 (Remarks to the Author):

The paper presents measurements of the 3rd harmonic of the ac susceptibility of a series of oxide superconductors. There is no distinction between real and imaginary part of this quantity. A broad precursor region above the critical temperature in a series of samples with significantly different T_c 's and coherence lengths was observed. In most materials the temperature dependence appears to be exponential. The samples have in common some form of spatial coordination of CuO₆ perovskite octahedra either in pure perovskite form or as layered Ruddlesden-Popper phases. It is rather amazing to see that this type of precursor effect in a rather easily accessible experimental quantity is only reported after more than 30 years of intensive research including a vast number of ac susceptibility measurements.

The observed effect is interpreted in terms of structural inhomogeneities. This is quantified by comparison with the derivative of T_c with uniaxial stress.

Unfortunately, another very important most prominent common feature, i.e. that the samples are oxides, is not considered. It is easy to understand that oxygen deficiency/excess is certainly kind of defect with a magnetic signature. The resulting doping/carrier concentration inhomogeneity might also account for a rather coherence-length independent effect. In this context, the neglect of any doping dependent features is hard to understand. In addition, there is no reason why other oxides containing Cu-O configurations other than octahedral perovskite coordination like YBCO are not considered. However, the most severe objection is of experimental nature. Even though the observed effect is called "unusual exponential temperature dependence of the diamagnetic response above T_c ", the response of the only really high- T_c material Hg1201 does not show this type of form at all. The corresponding curve in figure 1d is rounded on any scale. Ascribing an exponential decay coefficient to a banana is pretty audacious, and the rather small error margin of 0.5% in table 1 is only a result of cheating with a small temperature interval 1K above T_c . Looking closer, also the SRO curve of figure 1b is a "hanger" rather than a straight line.

Conclusion

The paper in its present form is not acceptable in any serious scientific journal. However, in view of the rather remarkable discovery of a (maybe exponential) broad precursor region in a series of oxides with significantly different T_c 's and coherence lengths, there might be a chance for a second try with a proper interpretation.

Some technical details:

1. The upper-case Greek " χ " is not shown in proper form in the ordinate labeling of figure 2.
2. The Vanadium curve of figure S1 is of miserable quality and should be replaced by the measurement of a better material.
3. In figure S2, calling the decrease of the specific heat exponential nicely underlines the author's lax understanding of exponential decay. In addition, describing a 40% difference in slope as somewhat steeper is pretty lazy, too. Such type of figures and interpretations should not appear in serious papers.

Reviewer #3 (Remarks to the Author):

This paper reports an observation of non-linear diamagnetism above T_c in four oxide superconductors with likely very different physical mechanisms for superconductivity. There is strong circumstantial evidence presented that this is induced by macroscopic strain-induced inhomogeneity.

The authors refreshingly put this out as a fascinating experimental correlation, and don't sully the paper with too much irrelevant theory, though possible directions are pointed out.

One question - I presume that this is diamagnetic, but I didn't see in the paper this stated directly.

I think this is a fascinating nugget of information, that will be of interest quite broadly, and will sponsor more thinking about inhomogeneous superconductivity - something quite difficult to arrange in BCS-like models with long-range interactions. I would favor publication.

Reviewer 1

We thank the Reviewer for the helpful comments and suggestions, especially on the scaling of the data, which indeed reinforces the universality that we have uncovered.

The manuscript by D. Pelc et al. reports an universal diamagnetic signature observed above the superconducting transition temperature (T_c) of various transition-metal oxide superconductors, namely SrTiO₃ (STO), Sr₂RuO₄ (SRO), La_{1.86}Sr_{0.14}CuO₄ (LSCO) and HgBa₂CuO₄ (Hg1201). Measurements of the third-harmonic magnetic response (M_3) to an applied field (H) in the kHz regime, made on both as-grown and uniaxially-strained single crystals, showed a predominantly exponential dependence of M_3 on temperature (T), in contrast to a power-law behavior that is expected from Ginzburg-Landau theory for conventional superconductors.

This work has both technical novelty and crucial significance for the study of superconducting fluctuations above the bulk T_c in oxide superconductors, where the superconducting energy gap is known to vary locally due to both chemical and structural inhomogeneities, the latter being intrinsic to oxide perovskites.

First, the present experiment is technically distinct from prior measurement of nonlinear electromagnetic response made in terms of conductivity at radio frequencies (Ref. 8). To my best knowledge, the simultaneous measurement of M_3 and sample strain reported here are the first of its kind. Importantly, by also performing control experiments on the conventional superconductors V, Pb, SnAg and Nb, which expectedly show power-law behavior, the technique used and thus the unconventional behavior observed in STO, SRO, LSCO and Hg1201 are properly validated.

Second, the fact that the magnitude of M_3 vs. T varies with uniaxial strain, while the exponential behavior remains unchanged, crucially connects this unconventional behavior to the presence of structural inhomogeneities. The appearance of a common ratio, of the exponential temperature scale to dT_c/dP_i (from Eqn. 2), is strong evidence for a universal pre-pairing scale (in Fig. 2b) rooted in local defects. These observations also suggest that strain engineering can be effectively used to manipulate local inhomogeneities as a tuning parameter of the superconducting fluctuations in these oxides.

In my opinion, the technical and scientific merits of this work amply justify its publication in Nature Communications. The manuscript is well written, the data clearly presented and the key results are of both general interest and timely importance. Listed below are some questions of curiosity that the authors may consider addressing, along with some constructive suggestions for further improving the manuscript:

- 1) Since the exponential behavior of M_3 vs. T is attributable to spatial inhomogeneity of the superconducting energy gap (Δ_0), it may be elucidating to plot the M_3 data vs. a "reduced energy" scale rather than a "reduced temperature" scale. Such a plot could perhaps allow all three panels of Fig.1 to collapse into one master plot, thus further strengthening the universality. In this regard, it is worth noting that the $2\Delta_0/kBT_c$ ratio for cuprate superconductors tends to deviate from mean-field value, as T_c is increased (J. Wei et al., Phys. Rev. B 57, 3650 (1998)). It

is also of interest to note the difference between “pairing” and “condensation” energies, with the latter reflecting T_c , in relation to the pseudogap phenomenon (G. Deutscher, Nature 397, 410 (1999)).

We have included a scaling plot in the revised manuscript (now Fig. 2a), which indeed reinforces the universality of the observed behavior. We agree that T_c is not necessarily the best parameter to use as an energy scale, especially in cuprates; the primary motivation for the T_c -dependence plot was to show the range of the characteristic scale Ξ for the different compounds, without an attempt to extract any quantitative relation. We thank the referee for the suggestion.

2) The strain hysteresis shown in Fig. 3 suggests that M3 vs. T could also be hysteretic, as one may expect from irreversible inhomogeneities associated with the plastic deformation. It may be worthwhile to elaborate on this point in the manuscript. In particular, would the occurrence of the latter hysteresis provide further evidence for strain-induced inhomogeneities?

We have not noticed any hysteresis in the magnetic response above T_c , which is perhaps to be expected, since the plastic deformation is performed at 300 K and the defects are ‘frozen in’ at low temperatures. In ongoing follow-up experiments, which will eventually be published as part of a separate study, we are testing the possibility that high-temperature annealing of stressed samples can return M3 to the pre-stressed condition.

3) In Fig. 1a, it would be clarifying to indicate the orientations of the crystal and of the magnetic field. For the latter, it would also help to quantify whether and how demagnetization factors, due to sample shape and field orientation, may affect the M3 vs. T data.

We have added the orientations to the caption of Fig. 1. We have not considered demagnetization factors because the measurements are nominally above T_c , where the linear susceptibility is small and the local field is not significantly distorted by the diamagnetism. However, all samples are of rectangular shape, with non-negligible thickness (typical aspect ratio 1:4), and thus we estimate that demagnetization effects would lead to no more than a 50% increase of the local field below T_c .

4) Finally, the title of the manuscript could be made a little less abstract, particularly for the general reader who may not be familiar with the phenomenological complexities of precursor superconductivity in transition-metal perovskites.

We agree, and have changed the title to “*Universal superconducting precursor in cuprate, ruthenate and titanate superconductors*”

Reviewer 2

We thank the Reviewer for the constructive comments, which have prompted us to perform further measurements and to modify the discussion.

The paper presents measurements of the 3rd harmonic of the ac susceptibility of a series of oxide superconductors. There is no distinction between real and imaginary part of this quantity. A broad precursor region above the critical temperature in a series of samples with significantly different T_c 's and coherence lengths was observed. In most materials the temperature dependence appears to be exponential. The samples have in common some form of spatial coordination of CuO₆ perovskite octahedra either in pure perovskite form or as layered Ruddlesden-Popper phases. It is rather amazing to see that this type of precursor effect in a rather easily accessible experimental quantity is only reported after more than 30 years of intensive research including a vast number of ac susceptibility measurements.

The observed effect is interpreted in terms of structural inhomogeneities. This is quantified by comparison with the derivative of T_c with uniaxial stress. Unfortunately, another very important most prominent common feature, i.e. that the samples are oxides, is not considered. It is easy to understand that oxygen deficiency/excess is certainly kind of defect with a magnetic signature. The resulting doping/carrier concentration inhomogeneity might also account for a rather coherence-length independent effect. In this context, the neglect of any doping dependent features is hard to understand.

We agree that perovskite-based oxides are especially prone to oxygen non-stoichiometry, and have added a statement on this in the revised manuscript. It is, however, not clear how plastic deformation could influence the non-stoichiometry, and we therefore consider this a less likely origin of the inhomogeneity that we discuss. Yet there might well be a complex interplay of oxygen and structural defects, which is at this point purely speculative (and now included in the manuscript as such). In the cuprates, the doping dependence of the precursor was been studied in detail with complementary experimental probes in refs. 8, 20 and 21, where it was shown that there is no significant doping dependence; this strongly suggests that doping-related disorder plays a minor role at best. In addition, the doping level of STO is very small, and SRO is nominally undoped, but these two oxides nevertheless exhibit a similar response above T_c . In contrast, perovskite-structure-related inhomogeneity would naturally have rather small doping dependence, and this is one of the reasons why it is our preferred explanation. Yet we note that, at this time, we do not have microscopic insight, and we have thus strived to keep speculation within reasonable bounds.

In addition, there is no reason why other oxides containing Cu-O configurations other than octahedral perovskite coordination like YBCO are not considered.

The goal of our study was to cast a wide net in terms of distinct families of oxide superconductors, not to investigate any particular family in detail. An additional reason is of purely practical nature, i.e., sample availability. We note, however, that whereas YBCO does not feature octahedral coordination (but instead is missing one of the apical oxygen atoms), its structure is still perovskite-derived and prone to similar instabilities. Notably, similar exponential

behaviour is also found in the nonlinear conductivity and paraconductivity of YBCO (see e.g. PRB 84, 014522 (2011) and ref. 8).

However, the most severe objection is of experimental nature. Even though the observed effect is called “unusual exponential temperature dependence of the diamagnetic response above T_c ”, the response of the only really high- T_c material Hg1201 does not show this type of form at all. The corresponding curve in figure 1d is rounded on any scale. Ascribing an exponential decay coefficient to a banana is pretty audacious, and the rather small error margin of 0.5% in table 1 is only a result of cheating with a small temperature interval 1K above T_c . Looking closer, also the SRO curve of figure 1b is a “hanger” rather than a straight line.

We disagree with the assertion that the Hg1201 data do not show an exponential form “at all”; the data on a log-linear plot fairly closely follow a straight line over more than an one order of magnitude (see the new Fig. S4b, where this is highlighted by the overlaid line). To make the argument more quantitative, we now include reduced sums of squared residuals of linear fits to $\log M_3$ in Table 1, which are small for all materials. Notably, a fit to, e.g., a power law temperature dependence (with a free power) gives roughly three times higher residual sums. Within 1 K, the signal decays by roughly a factor of 20 (i.e., the dynamic range is high), so a fit in this range should not be termed “cheating”. We also point to the fact that the exponential precursor was observed and characterized in cuprates, including Hg1201, using multiple experimental probes (linear and nonlinear conductivity, torque magnetometry, etc. – see refs. 8, 20 and 21), so it is beyond reasonable doubt that the response in Hg1201 is indeed consistent with this simple behavior.

However, to further strengthen our point, we have performed additional measurements on several Hg1201 samples, which are now shown in Fig. 1d and Fig. S4. The measurements also demonstrate the sample-to-sample variability that is typically obtained for Hg1201, and the mean slope of the exponential is slightly smaller than determined in the original version of the manuscript. We note that the carrier doping process in Hg1201 involves oxygen diffusion, and it is not uncommon to observe surface superconductivity at higher temperatures than the bulk. Importantly, nonlinear magnetization is an extremely sensitive probe of such effects, which cause ‘bumps’ above the bulk T_c ; it is a significantly better probe of sample quality than linear response. Yet, for that reason, the tails may depart from a pure exponential, and sample-to-sample variability may be larger than in the substitutionally-doped LSCO.

To further highlight the robust universality of the observed response, we now include a scaling plot in the revised manuscript (new Fig. 2a) that shows data for all investigated materials. The resulting master curve is highly consistent with simple predominant exponential decay.

Conclusion

The paper in its present form is not acceptable in any serious scientific journal. However, in view of the rather remarkable discovery of a (maybe exponential) broad precursor region in a series of oxides with significantly different T_c 's and coherence lengths, there might be a chance for a second try with a proper interpretation.

We have added new data, a scaling plot, and widened the discussion of possible origins of the observed behavior, in line with the Reviewer's suggestions.

Some technical details:

1. The upper-case Greek "Xi" is not shown in proper form in the ordinate labeling of figure 2.

This is now corrected.

2. The Vanadium curve of figure S1 is of miserable quality and should be replaced by the measurement of a better material.

We have replaced the vanadium curve with a new measurement on indium.

3. In figure S2, calling the decrease of the specific heat exponential nicely underlines the author's lax understanding of exponential decay. In addition, describing a 40% difference in slope as somewhat steeper is pretty lazy, too. Such type of figures and interpretations should not appear in serious papers.

We have changed the language pertaining to Fig. S2, but note that the specific heat data are indeed consistent with an exponential decay (within the somewhat limited data quality) over nearly one decade, as highlighted by the straight line.

Reviewer 3

We are grateful for the Reviewer's positive comments, including the recognition of our (conscious) attempt to minimize theoretical speculation.

This paper reports an observation of non-linear diamagnetism above T_c in four oxide superconductors with likely very different physical mechanisms for superconductivity. There is strong circumstantial evidence presented that this is induced by macroscopic strain-induced inhomogeneity.

The authors refreshingly put this out as a fascinating experimental correlation, and don't sully the paper with too much irrelevant theory, though possible directions are pointed out.

One question - I presume that this is diamagnetic, but I didn't see in the paper this stated directly.

Yes, the signal is indeed diamagnetic (this is stated in the abstract and on page 2).

I think this is a fascinating nugget of information, that will be of interest quite broadly, and will sponsor more thinking about inhomogeneous superconductivity - something quite difficult to arrange in BCS-like models with long-range interactions. I would favor publication.

REVIEWERS' COMMENTS:

Reviewer #1 (Remarks to the Author):

The author's rebuttal and revisions more than satisfy the comments of my first report. In particular, the masterplot in Fig. 2(a) now firmly demonstrates the observed universality, which the new title eloquently articulates. I have no doubt that publication of this paper in Nature Communications will significantly advance fresh inquiry on the important topic of precursor pairing in [transition-metal oxide] superconductors.

Reviewer #2 (Remarks to the Author):

As the authors did account for most of the reviewer's critics/suggestions I recommend publication in Nature Communications.